# Mechanistic Insights into Substrate Recognition of Human Nucleoside Diphosphate Kinase C Based on Nucleotide-Induced Structural Changes

**DOI:** 10.3390/ijms25189768

**Published:** 2024-09-10

**Authors:** Rezan Amjadi, Sebastiaan Werten, Santosh Kumar Lomada, Clara Baldin, Klaus Scheffzek, Theresia Dunzendorfer-Matt, Thomas Wieland

**Affiliations:** 1Institute of Molecular Biochemistry, Medical University of Innsbruck, Innrain 80/82, 6020 Innsbruck, Austria; rezan.amjadi@i-med.ac.at (R.A.); klaus.scheffzek@i-med.ac.at (K.S.); 2Department of General, Inorganic and Theoretical Chemistry, University of Innsbruck, Innrain 80/82, 6020 Innsbruck, Austria; sebastiaan.werten@uibk.ac.at; 3Experimental Pharmacology Mannheim, European Center for Angioscience, Medical Faculty Mannheim, Heidelberg University, Ludolf-Krehl-Str. 13–17, 68167 Mannheim, Germany; santosh.lomada@medma.uni-heidelberg.de; 4Department of Microbiology, University of Innsbruck, Technikerstraße 25, 6020 Innsbruck, Austria; clara.baldin@uibk.ac.at; 5DZHK (German Center for Cardiovascular Research), Partner Site Heidelberg/Mannheim, 68167 Mannheim, Germany

**Keywords:** nucleoside diphosphate kinase C, *nme3*, enzymatic mechanism, magnesium binding, X-ray structure

## Abstract

Nucleoside diphosphate kinases (NDPKs) are encoded by *nme* genes and exist in various isoforms. Based on interactions with other proteins, they are involved in signal transduction, development and pathological processes such as tumorigenesis, metastasis and heart failure. In this study, we report a 1.25 Å resolution structure of human homohexameric NDPK-C bound to ADP and describe the yet unknown complexes formed with GDP, UDP and cAMP, all obtained at a high resolution via X-ray crystallography. Each nucleotide represents a distinct group of mono- or diphosphate purine or pyrimidine bases. We analyzed different NDPK-C nucleotide complexes in the presence and absence of Mg^2+^ and explain how this ion plays an essential role in NDPKs’ phosphotransferase activity. By analyzing a nucleotide-depleted NDPK-C structure, we detected conformational changes upon substrate binding and identify flexible regions in the substrate binding site. A comparison of NDPK-C with other human isoforms revealed a strong similarity in the overall composition with regard to the 3D structure, but significant differences in the charge and hydrophobicity of the isoforms’ surfaces. This may play a role in isoform-specific NDPK interactions with ligands and/or important complex partners like other NDPK isoforms, as well as monomeric and heterotrimeric G proteins. Considering the recently discovered role of NDPK-C in different pathologies, these high-resolution structures thus might provide a basis for interaction studies with other proteins or small ligands, like activators or inhibitors.

## 1. Introduction

The enzyme family of nucleoside diphosphate kinases (NDPKs) is involved in multiple physiological and, more importantly, pathological processes. NDPKs contribute to the development of pathologies as diverse as tumorigenesis, metastasis, retinopathy and severe cardiac dysfunction. In many cases, their underlying molecular mechanisms remain elusive [1,2]. With regard to their common enzymatic activity, NDPKs are NTP/NDP transphosphorylases encoded by the *nme* genes. They catalyze the transfer of the γ-phosphate group from an NTP to an NDP via the formation of a high-energy intermediate on a catalytic histidine residue [3,4,5,6,7]. Thus, these enzymes provide all di- and triphosphate nucleosides and deoxynucleosides containing natural purine or pyrimidine with ATP as the main intracellular phosphate donor [8,9,10,11]. Apart from their housekeeping role, NDPKs are involved in many regulatory functions in different cellular processes, such as the proliferation and regulation of endocytosis [12,13,14,15], apoptosis [16,17], GTP supply and the activation of heterotrimeric G proteins [18,19,20,21], GTP supply to dynamin and the control of gene expression [22,23]. All these different functions most likely require specific subcellular localization and the differential oligomerization of NDPK isoforms [2,24].

The NDPK family members are structurally highly conserved from bacteria to humans and all eukaryotic and archaebacterial NDPKs are hexameric. Most prokaryotic NDPKs are tetramers, however. Functional oligomers are always constructed from a dimer formed by two monomer subunits assembled from head to tail. The enzymatic active and physiological functional hexameric NDPKs are built of three dimers and have subunits comprising approximately 150 residues which fold into ferredoxin-like structures [8,25,26]. Each subunit is made up of a four-stranded antiparallel β-sheet, connecting helices, a small compact structure termed the “Killer of prune” (Kpn)-loop, and N- and C-terminal segments. The packing of the α-helices around the β-sheets creates a very common structural motif called an αβ sandwich or βαββαβ fold [27]. The head-to-tail assembled dimers are the basic unit of the respective NDPK. The C-terminal segment, a highly flexible and variable area of NDPKs, guides trimerization to form the catalytic active hexamer [25,26,28]. The majority of residues that form the dimer interface are conserved from bacteria to humans and locate in helix α_1_, strand β_2_ and at the end of the C-terminal segment, whereas the trimer interface abundantly involves residues in the *Kpn*-loop [29].

Phylogenetic studies suggest that the mammalian NDPK family members cluster in two groups based on their domain organization. The protein products of *nme1* to *nme4* belong to group I, which are characterized by a single NDPK domain, while the proteins encoded by *nme5* to *nme10* form group II and harbor additional domains. The latter do not exhibit NDPK activity, except for the NME6 protein [30,31]. Among various NDPK family members encoded by ten human *nme* group I genes, NDPK-A and -B (protein products of the *nme1* and *nme2* genes, respectively) are the most abundant isoforms in humans and share 88% of their sequence identity [30]. The reduced expression of NDPK-A and -B is correlated with a higher “metastatic potential” in human cancer cells [32]. Recent data have revealed that the diminished endocytosis of integrins based on a reduced GTP supply to dynamin is a possible underlying mechanism [13]. Both variants can assemble to form homo- and heterohexamers [33,34]. However, in vivo, both NDPK-A and -B are present as hetero-hexamers, and the stoichiometry within the hexamers depends on the expression level of each NDPK isoform [35]. Given that NDPKs isolated from mammals are primarily hetero-hexamers, the constitution of such hetero-hexamers may have a significant impact on specific biological functions and how they interact with other proteins or substrates [28,35]. 

In contrast to the widely studied isoforms NDPK-A and -B as well as mitochondrial specialized NDPK-D [36,37], knowledge on the isoform NDPK-C, encoded by the *nme3* gene, is rather sparse. It is highly enriched in the mitochondrial outer membrane as well as in the membranes of the Golgi apparatus and the endoplasmic reticulum [30]. In the mitochondrial outer membrane, its catalytic activity in mitochondria fusion or fission was discovered to have an important role, thereby contributing to different pathologies such as ischemia/reperfusion-induced infarction, abnormalities in cerebellar function and mitochondrial neurodegenerative disorder [38,39]. Other recent data have shown that NDPK-C is upregulated in human heart failure and targeted to the plasma membrane and caveolae [18,19]. The 17 N-terminal hydrophobic residues of NDPK-C, which are likely used to anchor the protein to the membranes, are the main difference between NDPK-C and NDPK-A and -B [40,41]. While NDPK-C is approximately 65% identical to isoforms A and B on the primary sequence level, it shows a higher stability against denaturants and is more thermostable [42]. Most interestingly, NDPK-C was found to be essential for the complex formation of NDPK-B/NDPK-C hetero-oligomers with the heterotrimeric G proteins G_s_ and G_i_, securing GTP supply and G protein activation [19]. The upregulation of NDPK-C in heart failure induced an enhanced complex formation with G_i_, resulting in the increased chronic activation of this G protein, a reduction in cAMP production and thus a decrease in contractility in cardiomyocytes [19]. All these data point to NDPK-C as an interesting target for small-molecule inhibitors, the identification of which would need detailed structural information on this isoform. Beyond a single PDB deposition (1ZSG), a detailed structural interpretation of NDPK-C is, however, still missing. Here, we report the structure of NDPK-C bound to various nucleotides at up to a 1.25 Å resolution. In order to identify the conformational changes in NDPK-C upon substrate binding, we have analyzed NDPK-C individually in complex with ADP, GDP, UDP and cAMP and in the absence of nucleotides. The obtained details of the structural changes upon substrate binding will additionally help to understand the catalytic mechanism more accurately and open the path for structure-supported drug design.

## 2. Results and Discussion

In order to optimize recombinant human NDPK-C solubility and stability for crystallization, we removed the 17 residue amino-terminal anchor sequence. We produced a truncated version of human NDPK isoform C (residues Thr18–Glu169, NDPK-C hereafter) and determined the 3D structure via X-ray crystallography in the absence and presence of various nucleotides. A data analysis revealed a hexameric arrangement, which is organized as three dimers, as previously reported for other functional eukaryotic NDPK variants. Each of these three subunits are oriented with threefold rotational symmetry on top of each other, and the subunits are colored in blue, green and orange (Figure 1A). The protomers that form a dimer unit within the hexamer become visible upon a 90 degree rotation around the horizontal axis and are shown in light blue, light green and light orange (Figure 1B). The 156 residue spanning the NDPK-C subunit forms a four-stranded antiparallel β-sheet and two connecting α-helices to make a highly conserved hydrophobic core (Figure 1C). Like in other NDPKs’ structures, the hydrophobic core, which is assembled with β4β1β3β2 topology, is surrounded by α-helices and the *Kpn*-loop. The active site of NDPK-C is formed by residues located at the *Kpn*-loop, the helices α_2_-α_A_ and the strand β_4_. Regardless of the basic unit of oligomerization, each subunit inside the hexamer is involved in oligomer formation via identical structural elements. One of these key features is the *Kpn*-loop located in the area between the subunits to form the trimer interface, which also involves the helices α_1_ and α_3_ and the C-terminus (Figure 1C). Interactions between two subunits that are opposite each other lead to dimer formation. The dimer interface is stabilized via the β-augmentation of two β2 strands, one from each protomer. The helices α1 and α0 from each subunit are also part of the dimer interface (Figure 1D). Additionally, salt bridges and hydrogen bonds stabilize the dimer module (Figure 1E).

In order to confirm the oligomeric state of recombinant NDPK-C in solution, purified protein samples were analyzed using size exclusion chromatography in combination with refractive index and light scattering detection (SEC-RI-MALS). The data show a molecular mass of 103 kDa, which is in good agreement with the formation of a hexameric complex (calculated *M*_r_ = 105 kDa) (Figure 1F). By analyzing elution volumes corresponding to lower molecular masses, we could not detect any amounts of NDPK-C dimers or monomers. These data were supported by analyzing His-tagged NDPK-C directly after affinity purification by mass photometry, in which hexameric His-tagged NDPK-C was the predominant form (Appendix A). An analysis of purified NDPK-C protein using HPLC-based ion-pair chromatography revealed that the preparation did not contain any nucleotides co-purified from *E. coli* lysates. In order to determine the crystal structures of active NDPK-C in the presence of various substrates, we selected the two purine diphosphates ADP and GDP, the pyrimidine diphosphate UDP and the known competitive inhibitor 3′,5′-cyclic AMP (cAMP) [9,11,43]. The three-dimensional structures of human NDPK-C in complex with ADP, GDP, UDP and cAMP were determined at resolutions of 1.25 Å, 2.17 Å, 1.87 Å, and 1.77 Å, respectively (Table 1). Of note, we failed to generate NDPK-C crystals in the presence of a tri-phosphate substrate (ATP or GTP). 

### 2.1. The NDPK-C ADP Complex in the Absence and Presence of Mg^2+^

Crystals of NDPK-C in a complex with ADP grew in two different conditions: (1) in a metal chelating screening solution containing citrate (see the Methods section), we obtained crystals without the magnesium ion; and (2) under non-chelating crystallization conditions, a magnesium ion was detected in the active site. The presence or absence of Mg^2+^ did not influence the formation of the hexamer.

The magnesium-free ADP complex of human NDPK-C formed crystals belonging to space group *P*1 which diffracted at 1.25 Å. The aromatic adenine ring formed a π-stack with Phe77 of helix α_2_ and the sugar moiety associated with Val129 and Asn132 from the *Kpn*-loop (Figure 2A). The ribose was recognized via its C2’ and C3’ hydroxyl groups, which were close to Lys29 located in the linker between α_0_ and β_1_. The β-phosphate formed hydrogen bonds with Arg105 and Thr111. Additionally, the phosphate segment of the nucleotide was in contact with Arg122; the distance of its guanidine group to the β-phosphate of ADP was 2.9 Å (Figure 2A). The upper part of the nucleotide-binding pocket included indirect protein ligand contacts. The sugar and phosphate moieties were connected to Tyr69, His135 and Gly136 (Figure 2B) by three bridging water molecules.

The magnesium-containing crystals (belonging to space group *C*2) diffracted to a resolution of 2.10 Å. Essentially, we detected a very similar mode of nucleotide binding (Figure 2C) including the residues described above for the metal ion free form (Phe77, Val129, Asn132, Lys29, Arg105 and Thr111). However, a few details were different. Arg122 showed the same position and orientation, but its contact to the β-phosphate of ADP was lost as evidenced by a change in the distance from 2.9 to 4 Å (Figure 2C). A striking difference was observed in the upper part of the nucleotide-binding pocket. In addition to the residues already described (Tyr69, His135 and Gly136), Asp138 was also part of the pocket, and we observed four water molecules which localized in a square pyramidal manner around the metal ion; the fifth coordination site was occupied by an oxygen from the nucleotide α-phosphate (Figure 2D). The intermediary water molecules were in direct contact with each other, the ligand and the protein. Nevertheless, the overall conformation of the two structures with and without the metal ion was very similar (Appendix A). However, a superposition of the nucleotide-binding pockets in the absence and presence of magnesium revealed that the magnesium ion induces a change in the conformation of the phosphate groups with the consequence of a weakened connection of ADP to Arg122 (Figure 2E). Thus, the magnesium ion changed the number, localization and coordination of the water molecules as well as the conformation of the nucleotide phosphate chain. 

Previous structural studies use aluminum fluoride (AlF_3_) together with ADP and magnesium ions in order to mimic the transition state of the phosphate transfer reaction [44]. In the respective structure derived for the slime mold *Dictyostelium discoideum* (1KDN), AlF_3_ adopts a trigonal planar conformation and locates between the catalytic histidine residue and the β-phosphate of ADP where one would expect the γ-phosphate in case of ATP. Furthermore, AlF_3_ contacts Mg^2+^, His122 (N_δ1_), Lys16 (Nε), Tyr56 (aromatic oxygen), Arg96 (side chain) and Gly123 (backbone) (numbering according to the sequence of *D. discoideum*). In this structure, Mg^2+^ is octahedrally coordinated by three water molecules, the α- and the β-phosphate of ADP, and a fluorine atom of AlF_3_ mimicking the γ-phosphate [45]. In our NDPK-C structure, we observed the same coordination of Mg^2+^ to three water molecules and the α- and β-phosphate of ADP (Figure 2F).

### 2.2. Comparison of Human NDPK-C Ligated with Different Nucleotides

In order to see differences in substrate recognition between adenosine and guanosine nucleotides, we crystallized NDPK-C in the presence of GDP, cAMP or UDP. Crystals with GDP belonged to space group *P*2_1_, and the structure was solved at a resolution of 2.17 Å. For NDPK-C×GDP, we observed an unusual crystal packing: two hexamers contacted each other via Arg80 and GDP, sandwiching a SO_4_^2−^ ion from the crystallization screening solution (Figure 3A). A further interaction between Asn112 (helix α_2_) of one hexamer and Pro76 of the neighboring hexamer could be observed. The residues contacting the guanine nucleotide base are the same as in the ADP complex, except for Glu169, which makes contact specifically with the guanine base in the adjacent protomer (Figure 3B). The structure of NDPK-C bound to the pyrimidine nucleotide UDP-Mg^2+^ (solved at 1.85 Å) was essentially identical to the ADP-Mg^2+^ complex (Figure 3C).

We additionally crystallized the complex formed by human NDPK-C and 3´,5´-cyclic AMP (cAMP), which diffracted to 1.77 Å. Compared to the purine dinucleotides described above, cAMP showed a slightly different binding mode. Again, the base was stacked onto Phe77, but the hydrogen bond between the sugar moiety and the side chain of Lys29 disappeared; instead, Lys29 was in contact with the phosphate group. We also observed a difference in the lower part of the binding pocket: Arg105 and Arg122 together with the Gly136 backbone oxygen interacted indirectly with the phosphate group of cAMP via a water molecule (Figure 3D). In contrast to the non-cyclic nucleotides, the single phosphate of cAMP was in direct contact with Tyr69 (located in helix α_A_), and His135 (β_4_) compensated for the negative charge to stabilize the nucleotide. The superposition of NDPK-C×cAMP and NDPK-C×ADP complexes revealed that the helix hairpin α_A_-α_2_ and β-strand in the core of the monomer are involved in an interaction with cAMP, whereas the Kpn-loop has a more substantial impact on the interaction with the non-cyclic nucleotides (Figure 3E). Previous structural and biochemical studies have shown that cAMP can occupy the nucleotide-binding site of *Myxococcus xanthus* NDPK and thus inhibits the enzyme activity competitively [11]. It has also been shown earlier that cAMP interacts with human erythrocytic NDPK in a manner that is distinct from other nucleotides, thereby inhibiting the enzymatic activity [9,43]. The superposition of human homo-hexameric NDPK-C with bacterial NDPK showed that the binding sites and the interactions of the enzyme with cAMP were identical in both proteins, except for an isoleucine residue (Ile111 in *M. xanthus*) which corresponds to Val129 in human NDPK-C (Appendix A) [46]. In order to verify the proposed competitive inhibition by cAMP on NDPK-C activity, we measured the enzymatic activity of purified recombinant NDPK-C in the absence and presence of increasing concentrations of cAMP (1 and 5 mM). Unexpectedly, we detected no inhibition of human NDPK-C by cAMP, although only very low concentrations of standard nucleotides were used. These data suggest a rather low affinity of cAMP to NDPK-C (Figure 3F). 

Physiological free Mg^2+^ concentrations in the cytosol vary between 0.5 and 1 mM, or less than 5% of the total cellular Mg^2+^ content, due to the binding/buffering capacity of phospho-nucleotides, phosphometabolites and, potentially, proteins [47]. A previous report suggests that the first step in the phosphotransfer reaction, i.e., the transfer of phosphate from the nucleoside triphosphate (NTP) to the catalytic histidine residue in NDPK, can occur without Mg^2+^ and only the second step of the reaction (the transfer of phosphate to a different nucleoside diphosphate (NDP)) requires the presence of Mg^2+^ [46]. To analyze the turnover rate of GTP as well as the formation of ATP, we performed activity assays of NDPK-C at concentrations of free Mg^2+^ ranging from 20 nM to 5 mM as well as in the presence of a molar excess of EDTA, a potent chelator of divalent cations. We observed undiminished NDPK-C activity even at a very low Mg^2+^ concentration and no activity in the presence of EDTA (Figure 3F). We therefore analyzed the formation of the 1-phosphohistidine intermediate on His135 of NDPK-C using the monoclonal anti 1-phosphohistidine antibodies as described for NDPK-A and -B [48]. As shown in the inset of Figure 3F, the formation of a 1-phosphohistidine from ATP occurs only in the presence of Mg^2+^ but not in the presence of EDTA. First of all, these data indicate that we crystallized an active form of human NDPK-C in complex with ADP and Mg^2+^ in the *C*2 space group. Secondly, Mg^2+^ appears to be required in both steps of the phosphotransfer reaction, which from a mechanistic point of view, makes perfect sense. During catalysis, the Mg^2+^ is indispensable in stabilizing the negative charge of the leaving phosphate from ATP at the transition state. At physiologic pH, the phosphate groups of ATP are deprotonated; the accumulation of negative charges in the active site of the enzyme are shielded by positive charges of the residues forming the binding pocket, as well as by the magnesium cation. 

### 2.3. Structure of Nucleotide-Free Human NDPK-C

In order to investigate the impact of the nucleotide on the conformation of human NDPK-C, we tried to crystallize the enzyme in the absence of nucleotides. Since this was not successful, we depleted the nucleotide from NDPK-C×UDP crystals by back soaking. The resolution of the corresponding structure was reduced to 2.64 Å, consistent with the idea that the nucleotide stabilizes the protein. Interestingly, we observed that the area where the phosphate groups in the nucleotide complex are located was occupied by a phosphate ion from the crystallization liquid and even formed a hydrogen bond to the imidazole ring of the catalytic histidine (His135). The residues involved in the interaction with phosphate are distinct for each subunit, but mostly Lys29 and Tyr69 play a role in stabilizing the phosphate. Moreover, the phosphate interacts via water molecules with Arg105 and Gly136 of all subunits inside the NDPK-C hexamer (Figure 4A).

In the absence of a nucleotide, the α_2_ helix and the Kpn-loop no longer tightly close off the active site of NDPK-C. The distance between C_α_-Phe77 (one side of the gate) and C_α_-Val129 (the opposite side of the gate) is changed from 10.2 Å in the complex with the nucleotide to 11.3 Å in the absence of UDP at the binding site. The active site of NDPK-C may tend to be more open in the absence of a substrate (Figure 4B). We compared the B-factors of NDPK-C structures with and without UDP to measure flexibility in the macromolecule (Figure 4C). The removal of the UDP was accompanied by a small but detectable conformational change in the protein, particularly in the helices’ α_A_ (from residue Ala60 to Arg73) and α_2_ (from residue Pro75 to Ser87) region. The B-factor values in this region are markedly higher upon nucleotide depletion, reflecting an increase in flexibility.

We hypothesized that the position where we observe the phosphate ion in the nucleotide-free pocket could be the site occupied by the γ-phosphate in an ATP ligated NDPK-C. Since the phosphate electron density in our data set suggested a double conformation, we compared the electron density of apo-NDPK-C with histidine phosphorylated NDPK (1NSP), as well as with an NDPK × ADP-AlF_3_ complex from *D. discoideum* (1KDN). Superimposing the structure of human apo-NDPK-C to *D. discoideum-*treated NDPK × ADP × AlF_3_ revealed that the free phosphate essentially occupies the position of AlF_3_ right between the catalytic histidine and the β-phosphate of ADP in the same geometry as expected for the γ-phosphate of a nucleotide triphosphate (Figure 4D).

On the other hand, the structure of NDPK from *D. discoideum* demonstrated that the phosphate group leaving the phosphohistidine in the free active site forms a hydrogen bond to Tyr56 and two water molecules [4]. As mentioned above, the phosphate in the nucleotide-depleted structure of human NDPK-C partially occupies the nucleotide-binding pocket but has a longer distance to the catalytic histidine compared to the localization of phosphate in the structure of phosphorylated histidine. However, the observed interactions with side chains and with water molecules were the same (Figure 4D). Taken together, these data suggest that free phosphate in the active site of human NDPK-C likely mimics the phosphate in the transition state between the catalytic histidine and the γ-phosphate in the nucleoside triphosphate.

Recently, the remodeling of the binding site of an NDPK hexamer from *Mycobacterium tuberculosis* was reported, in which all the subunits of one trimer inside the hexamer show closed and opened conformations [49]. The remodeling is supported by evidence including (i) the completely unfolded α_A_; (ii) the appearance of a new conformation for helix α_2_; (iii) the increase in the β_2_ and β_3_ strands’ lengths; and (iv) the disappearance of the residues of the electron densities that connect helix α_A_ to helix α_2_. Therefore, in such an open conformation, the catalytic histidine at the bottom of the binding site is more accessible to other substrates [25,49], for example, histidine residues on the surface of other proteins. The isoform NDPK-B has been validated to be a *bona fide* mammalian protein histidine kinase, with at least three distinct substrates, the KCa3.1 and TRPV5 channels, and the G protein β subunit [48,50]. As the interaction of NDPK-B with heterotrimeric G proteins requires hetero-oligomerization with NDPK-C [19], we wondered whether such an open conformation would also exist for NDPK-C.

Therefore, we compared the overall conformation of NDPK-C in complex with NDP with the conformation of our nucleotide-depleted hexamer by superposing the main chain atoms. We did not obtain evidence for an unfolded α_A_ or a conformational change in helix α_2_. This might be due to a distinct structure of human NDPK-C. Most likely, however, the removal of the UDP from the already formed crystal did not really alter the packing and stabilization of these structural elements by crystal contacts. Nevertheless, the electron densities of the residues that form a linker between the α_A_ and α_2_ helices have disappeared in one trimer of apo NDPK-C. These data indicate that conformational changes in NDPK-C upon NDP removal can occur, with the helix α_2_ displaying the highest considerable shift in comparison with the nucleotide complexed protein. We therefore propose that the observed structural shift of the α_2_ helix in the absence of a bound nucleotide in NDPK-C indicates the possibility of adjusting the binding pocket to the uptake of other substrate molecules for this isoform as well. This should, however, be studied further and requires experimental proof that NDPK-C, like NDPK-B, can act as a protein histidine kinase.

### 2.4. Differences between the NDPK B and C Isoforms Might Contribute to Isoform-Specific Interactions

A variety of structures for NDPK from various species have been solved (for review see [25,26,28]. However, a high-resolution structure displaying details from the catalytic center for the human isoform C was still missing. Since the isoforms show differences in their expression profiles and interaction partners, we were interested in how the two B and C isoforms could be specifically addressed by their interaction partners or how potential heterohexamers formed between NDPK-B and -C could exhibit specific surfaces. A primary sequence alignment shows a high level of overall conservation (Figure 5A). Within the ~170 residues of the NDPK-C sequence, 12 residues show an isoform-specific conservation and a clustering close to the C-terminus. Only three of the isoform-specific conserved residues can be mapped to the surface of NDPK and may contribute to specific recognition by binding partners or inhibitors. In NDPK-B (PDB entry 1NUE), we observe a salt bridge between Lys143 and Asp148, which are close to the C-terminus (Figure 5B). In the NDPK-C primary sequence, these residues are replaced by Glu160 and His165, respectively, which are also located on the surface but do not form a salt bridge (Figure 5C). A conserved glycine residue (Gly63) locates close to the nucleotide-binding pocket in NDPK-B, and due to a lack of a carbon side chain the NDPK-B pocket, it seems to be more accessible at the side of the base contacts. In NDPK-C, the corresponding position is occupied by an arginine residue (Arg80). The bulky side chain closes the substrate pocket in addition to a positive charge on the surface. These differences in surface charge distribution are small but might contribute to specific interactions with other proteins or ligands. 

## 3. Materials and Methods

### 3.1. Cloning, Protein Preparation and Purification

A polymerase chain reaction (PCR) was performed using human NDPK-C cDNA [19] as a template together with the forward primer oligonucleotide 5′-d (GCTATTCATATGACTGGTGCTCATGAACGCACCTTTCTGGCCGTCAAACC)-3′ and reverse primer 5´-d (GGCCATTGGCTGTACGAGTAAGGATCCATTAGC)-3′. The forward primer introduces an *Nde*I site (CATATG, underlined) containing a start codon (AUG) followed by codons for Thr18-Lys29 residues of the human NDPK-C sequence. The reverse primer binds to Gly164–Glu169 codons and inserts a *Bam*HI site (GGATCC). The *Nde*I/*Bam*HI digested fragment was ligated into a pET-derived expression plasmid containing an N-terminal TEV-cleavable (His)_6_-tag sequence, predigested with *Nde*I and *Bam*HI. The final expression plasmid encodes a 173 amino acid polypeptide including an N-terminal His-tag followed by the N-terminal truncated NDPK-C sequence (res Thr 18–Glu 169 corresponding to UniProtID Q13232). 

The plasmid encoding His-NDPK-C was transformed into bacterial expression strain *E. coli* BL21 (DE3). Cultures were grown at 37 °C in Terrific broth medium supplemented with 50 µg/mL kanamycin. Protein expression was induced at a density of 0.5–0.6 by the addition of isopropyl thio-β-D-galactopyranoside (IPTG) at 1mM final concentration, and cultures were shifted to 16 °C and kept overnight. Bacteria were harvested by centrifugation at 11,000× *g* at 4 °C for 30 min and resuspended in lysis buffer (20 mM Tris-HCl (pH 8.0), 300 mM NaCl and 5 mM MgCl_2_). Samples were stored at −80 °C until use. His-tagged NDPK-C was purified by affinity chromatography on a HisTrap FastFlow cartridge (Cytiva) equilibrated in lysis buffer supplemented with 5% glycerol, 20 mM imidazole and 1 mM DTT. Elution was performed by loading buffer supplemented with 400 mM imidazole. The eluate containing NDPK-C was loaded to a size-exclusion chromatography column (HiLoad Superdex 200, Cytiva) equilibrated in lysis buffer supplemented with 2 mM DTT. The TEV cleavage reaction was set up as follows: protease was added to His-NDPK-C in a 1:10 molar ratio under slightly reducing conditions (1 mM DTT); and digestion was performed at room temperature overnight. A second size exclusion step was performed in the same buffer as above, except for the pH, which was set to 8.5, and untagged NDPK-C was separated from the reaction mixture. All purification steps were monitored by SDS-PAGE. The protein concentration was determined via UV absorbance at 280 nm, adjusted to 8 mg/mL (450 µM) before samples were snap-frozen and stored at −80 °C until use. 

### 3.2. Protein Characterization and Enzymatic Activity Assays

Purified NDPK-C samples were subjected to analytical size-exclusion chromatography combined with multi-angle light scattering and refractive index measurements (SEC-RI-MALS). Samples at various protein concentrations (60 to 600 µM) were injected into an analytical size-exclusion column (Superdex 200, 5/150, Cytiva) equilibrated in the same gel filtration buffer as above and eluted at a flow rate of 0.3 mL/min. RI measurements were performed at 25 °C.

Enzymatic assays were set up by mixing NDPK-C with nucleotide pairs (either ADP and GTP or GDP and ATP) in a 1: 2.5 × 10^5^ molar ratio (buffer with 20 nM Mg^2+^). Mixtures were incubated at 25 °C, samples were taken after several time points (10 s/20 s/40 s/60 s/90 s/160 s) and enzymatic reactions were stopped by thermal denaturation at 90 °C for 1 min. Cleared supernatants containing remaining or converted nucleotides were analyzed in duplicates by ion-pair chromatography using a reversed-phase C18 HPLC column in a mobile phase containing K_x_H_(3−x)_PO_4_ (pH 6.2), 35 mM tetrabutylammonium bromide and 7.5% (*v*/*v*) CH_3_CN under isocratic conditions. Experiments were performed in the absence and presence of potential inhibitors. For analysis of the magnesium dependence, we used NDPK-C at a final concentration of 20 nM and Mg^2+^-chelating EDTA at 1 mM final concentration (5×10^4^-fold molar excess of EDTA). For cAMP inhibition studies, we used 2 nM NDPK-C and cAMP at final concentrations of 1 mM or 5 mM (5 × 10^5^-fold or 2.5 × 10^6^-fold molar excess of cAMP). Standard solutions of the nucleotides were injected in serial dilutions at seven different concentrations in triplicates and used to quantify enzyme activity. Quantification of the nucleotides was performed via peak integration using Chromeleon software (7.2.10) against a standard curve. Graphs were created using the software GraphPad Prism (10.2.0).

The autophosphorylation of NDPK-C was analyzed with ATP as substrate in the presence of MgCl_2_. A total of 50 ng of NDPK-C was incubated with and without 2 mM MgCl_2_ for 5 min at room temperature in 30 µL of reaction buffer (20 mM Tris-HCl (pH 8.0), 150 mM NaCl, 1 mM EDTA and 1 mM ATP). The phosphorylation was stopped by adding 5 mM EDTA for chelation of free Mg^2+^. A total of 10 µL of 4× loading buffer (50% glycerol, 10% β-mercaptoethanol, 7.5% SDS, 300 mM Tris-HCl (pH 6.8) and 0.25% bromophenol blue) were added and proteins separated without heat denaturation on two 15% SDS-PAGE gels. The first gel was stained with SYPRO Ruby Protein Gel Stain (ThermoFisher Scientific, Waltham, MA, USA) according to the manufacturer’s instructions. Proteins from the second gel were transferred onto a 0.2 µm nitrocellulose membrane (Amersham Protran, Buckinghamshire, UK) and probed overnight at 4 °C with 1:1000 anti-N1-Phosphohistidine (anti-1p-His) antibody (clone SC1-1, Merck Millipore, Burlington, MA, USA), followed by horseradish peroxidase conjugated secondary antibody (ThermoFisher Scientific) in a 1:80,000 dilution at room temperature for 60 min. The blots were developed using SuperSignal West Femto Maximum Sensitivity Substrate (ThermoFisher Scientific) and visualized with the Vilber Fusion FX.

### 3.3. Crystallization

Nucleotides were dissolved in Tris buffer, and the pH was set to 7.4 before adding them to NDPK-C in at least twofold molar excess in order to saturate the enzyme. Since the crystallization efficiency of NDPK-C was strongly affected by the type and concentration of nucleotides used, optimal conditions were experimentally determined. Crystallization screening experiments were set up with NDPK-C (4 mg/mL) in the presence of either 1 mM ADP, GDP, cAMP or UDP (Sigma-Aldrich, St. Louis, MO, USA) in hanging drops under two different crystallization conditions at 291 K (18 °C). Conditions were optimized by varying the molar ratios between nucleotide and protein (Table 2). The crystals obtained in 6% PEG 3350, 150 mM Li_2_SO_4_ and 100 mM trisodium citrate (pH 5.4) were cryo-protected by adding 25% glycerol (*v*/*v*) to the mother liquor and were flash-frozen in liquid nitrogen. Crystals acquired in 27% glycerol, 7% PEG 8000 and 40 mM KH_2_PO_4_ were frozen in liquid nitrogen without extra cryo-protectant. Since the crystallization of nucleotide-free protein did not succeed, we used the soaking strategy to deplete the UDP from NDPK-C crystal in a complex with UDP. The optimized UDP complex crystals were soaked in 27% glycerol, 7% PEG 8000 and 40 mM KH_2_PO_4_ in the absence of UDP in solution with different soaking times. The electron density correlated with UDP was not detected in the crystals after 3 h of soaking time.

### 3.4. X-ray Data Collection

X-ray diffraction data were collected at 100 K at the ID23-1, ID23-2 and ID30A-3 beamlines [51,52,53] of the European Synchrotron Radiation Facility (ESRF), Grenoble, France. Data for 8QW3 and 8QVZ were processed using the programs iMosflm [54], Aimless and the CCP4 suite [55]. All other data sets were processed using XDS [56,57]. The structure of the ADP complex (8QW3) was solved by molecular replacement (MR) using Phaser [58] and PDB entry 1ZS6 as the search model. The final model was then used to solve the structures of the other complexes. All structures were refined using Coot [59] and Refmac [60,61]. Figures were generated using PyMOL (v 2.5.4.).

## 4. Conclusions

Here, the conformation of human NDPK-C in complex with various nucleotides was studied. The superposition of all main-chain atoms in the nucleotide-binding site showed that most of the residues involved in the interaction with nucleotides were identical. However, the protein complex with cAMP showed distinct interactions in the active site of NDPK-C, which overall did not lead to conformational changes in the protein in the presence of cAMP compared with other nucleotides in this study. The appearance of Mg^2+^ ions in all subunits inside the hexamer suggest that the cation interacts with α- and β-phosphates. The space where the γ-phosphate would be is occupied by a water molecule, which provides the sixth coordination of Mg^2+^. 

In addition, we studied the possibility of conformational changes due to the absence of the nucleotide from the active site of NDPK-C. Upon nucleotide depletion from NDPK-C, a PO_4_^3−^ ion from the crystallization solution localized close to the catalytic histidine at a position where one would expect the γ-phosphate of a triphosphate nucleotide in the transition state. The conformation of secondary structure elements did not show obvious changes except for the α_A_ and α_2_ helices. Nevertheless, the cleft of the nucleotide-binding site covered by the α_A_-α_2_ region appeared to be more open in the absence of nucleotides. This could indicate that the cleft may harbor substrates other than nucleotides. It is becoming increasingly apparent that the heterooligomerization of different NDPK isoforms is required for subcellular localization and specific functions. Since no such heterooligomers have been structurally studied so far, the detailed structures of NDPK-C presented here in the absence and presence of different bound nucleotides will help to understand differences in substrate recognition and facilitate predictions regarding interactions with other proteins. Considering further the recently discovered roles of the catalytic activity of NDPK-C in different pathologies [19,38,39], these structures will be substantial for developing small-molecule activators or inhibitors.

## Figures and Tables

**Figure 1 ijms-25-09768-f001:**
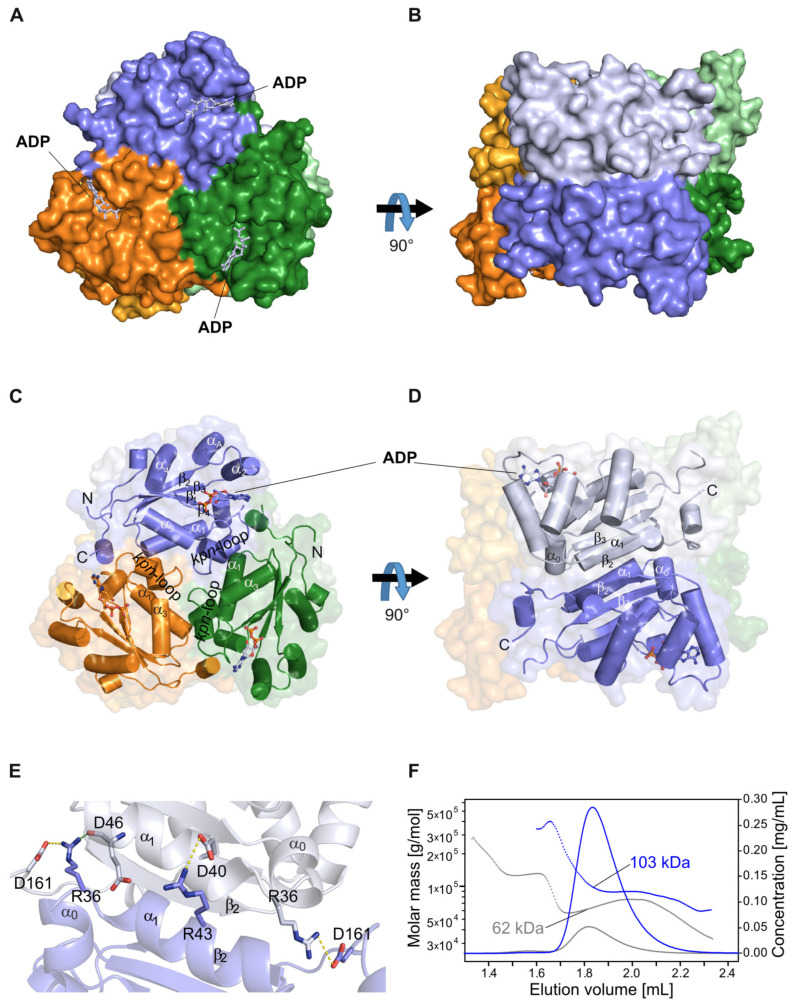
Overall view of human NDPK-C in complex with ADP—The homo-hexameric structure of NDPK-C can be considered as a trimer of dimers. (**A**) Top view—the mainly visible subunits oriented with threefold rotational symmetry are colored in blue, green and orange, respectively; the nucleotide ADP is shown in light blue. (**B**) Side view—generated by a 90 degree rotation around the horizontal axis, with same colors as in A. Here, the corresponding dimer-forming protomers are visible and colored in light blue, light green and light orange, respectively. (**C**) Top view—with all assigned secondary structure elements indicated and labeled in protomer 1 (blue) as in A. Only elements involved in the trimer contacts between the basic dimers are labeled in protomers 2 and 3 (green and orange). (**D**) Side view (orientation as in B) indicating the elements forming the dimer interface. (**E**) Zoomed-in view of the dimer interface, which is formed by an augmentation via β3 strands and additionally stabilized by salt bridges and hydrogen bonds (shown in dashed lines). (**F**) Analysis of the NDPK-C oligomeric state in solution. Chromatogram of an analytical native gel filtration analysis with refractive index (RI) and multiple-angle light scattering (MALS) detection. The calculated molar masses (left axis) are shown as dotted lines for NDPK-C (blue) and a control protein (Bovine serum albumin, shown in grey). Masses corresponding to concentration peak maxima (RI based solid lines, right axis) are displayed.

**Figure 2 ijms-25-09768-f002:**
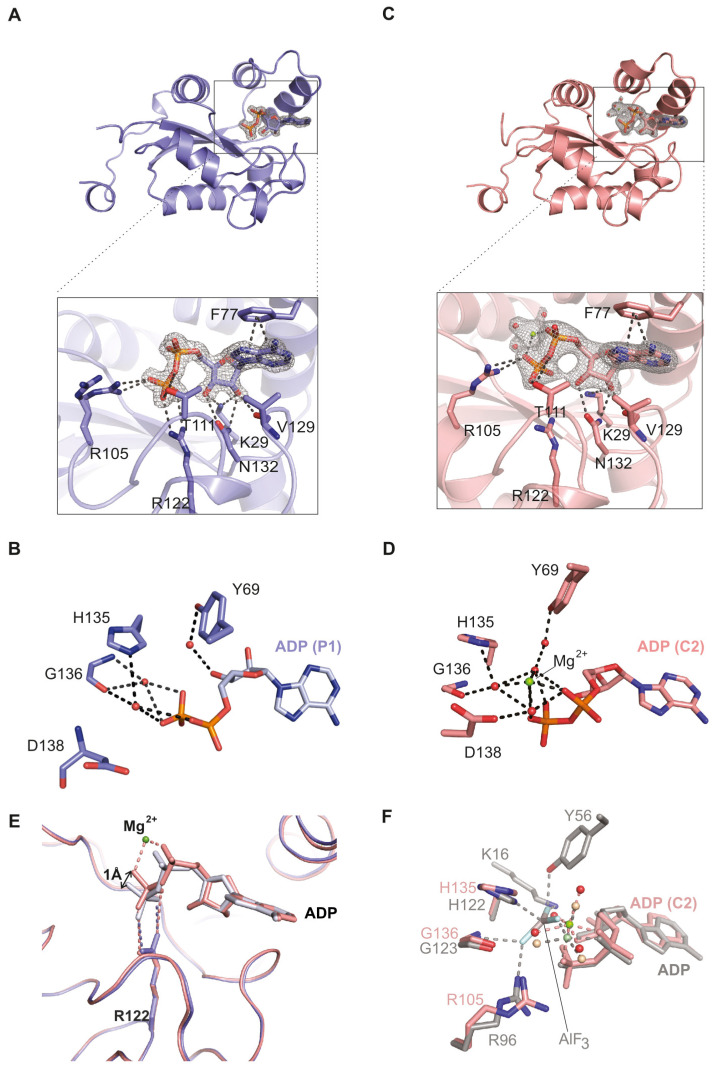
Human NDPK-C×ADP structure in the absence and presence of Mg^2+^—(**A**) the ribbon model (blue) shows protomer 1 of the magnesium-free hexameric NDPK-C structure in complex with ADP (space group *P*1). The Fo-Fc electron density map of the substrate contoured at σ = 1.3 is shown as a gray mesh together with the modeled ADP in atomic coloring. The nucleotide-binding site (boxed) is shown in a close-up view below. Lys29, Phe77, Arg105, Arg122, Val129 and Asn132 form the ADP-binding pocket (3 Å radius). (**B**) Stabilization of ADP in the active site of NDPK-C via indirect interaction with three water molecules (red spheres) bridging Tyr69, His135 and Gly136. (**C**) The ribbon model (pink) shows one NDPK-C subunit in the presence of magnesium (space group *C*2). The corresponding Fo-Fc map contoured at σ = 1.2 includes additional density that could be assigned to a magnesium ion and three water molecules (green and red spheres, respectively). The distance of Arg122 is longer than that in A (4 Å). (**D**) ADP stabilization in the presence of Mg^2+^ (green sphere) also includes Asp138, which is close to the β-phosphate and one of the four water molecules (red spheres). (**E**) Superposition of the two nucleotide pockets (A and C) show a change in the conformation of the phosphate chain of ADP upon metal ion binding. The black arrow indicates the movement of the β-phosphate toward the Mg^2+^ ion (with a respective distance of 1 Å) and the weakening of its interaction with Arg122. (**F**) Superposition of human NDPK-C (pink-and-green Mg^2+^ and four water molecules in red) with a *D. discoideum* NDPK-ADP-AlF_3_ complex (1KDN in grey with light-green Mg^2+^ and light-brown water molecules) showing the metal ion in octahedral coordination.

**Figure 3 ijms-25-09768-f003:**
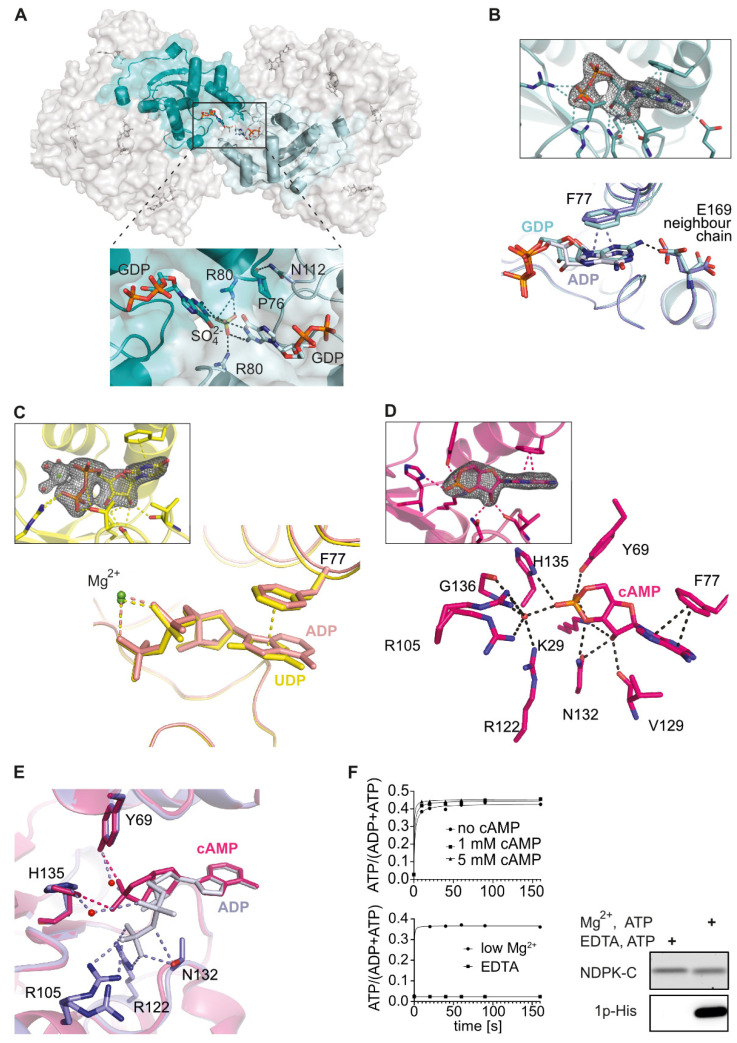
NDPK-C binding mode for different nucleotides (**A**) Two hexamers of NDPK-C in complex with GDP are linked via a bridging SO_4_^2−^ between subunits from each hexamer. The magnification view (below) shows the interface of two subunits. We observed contacts between Arg80 from each hexamer and Pro76 (cyan) with Asn112 from the other hexamer (light cyan). (**B**) Electron density map for the ligand GDP and superposition of NDPK-C×ADP (blue) with NDPK-C×GDP (cyan) complexes. In the GDP complex, the C2 amino group of the guanine base interacts with Glu169 from the neighboring chain inside the hexamer. (**C**) Electron density map for the ligand UDP and superposition of NDPK-C×UDP on NDPK-C×ADP (space group *C*2 with magnesium) shows that Phe77 shows a similar π-stack with purine and pyrimidine bases. (**D**) Electron density map for the ligand cAMP and active site of human NDPK-C in complex with cAMP. Lys29, Tyr69, Phe77, Val129, Asn132 and His135 directly interact with cAMP, while Arg105, Arg122 and Gly136 are involved in interaction with cAMP via a water molecule (shown as red sphere). (**E**) Superposition of NDPK-C×cAMP with NDPK-C×ADP complexes shows the different orientation and interactions of cAMP (pink) with active site residues compared to ADP (blue). In contrast to the ADP coordination the distance between the α-phosphate of cAMP and the Kpn-loop is larger, and this phosphate is closer to Tyr69 and the catalytically active His135. (**F**) NDPK-C activity in vitro. Upper panel: Enzymatic activity of recombinant NDPK-C (2 nM) was analyzed in the absence (spheres) or presence of cAMP at 1 mM (squares) or 5 mM (triangles); samples were taken at the indicated time points. Formation of ATP from ADP and GTP was analyzed by quantitative ion-pair chromatography (HPLC). Lower panel: Similar experiments were performed at low Mg^2+^ concentration (20 nM, spheres) or in the presence of 5 mM EDTA (squares); samples were taken at the indicated time points. Formation of GDP and ATP was analyzed by quantitative ion-pair chromatography (HPLC); data are shown for ATP accumulation. Right: Purified NDPK-C visualized with SYPRO Ruby Protein Gel Stain (upper panel) and the autophosphorylation of the catalytic histidine residue in the presence of Mg^2+^-ATP as detected with the anti-1p-His-antibody (lower panel).

**Figure 4 ijms-25-09768-f004:**
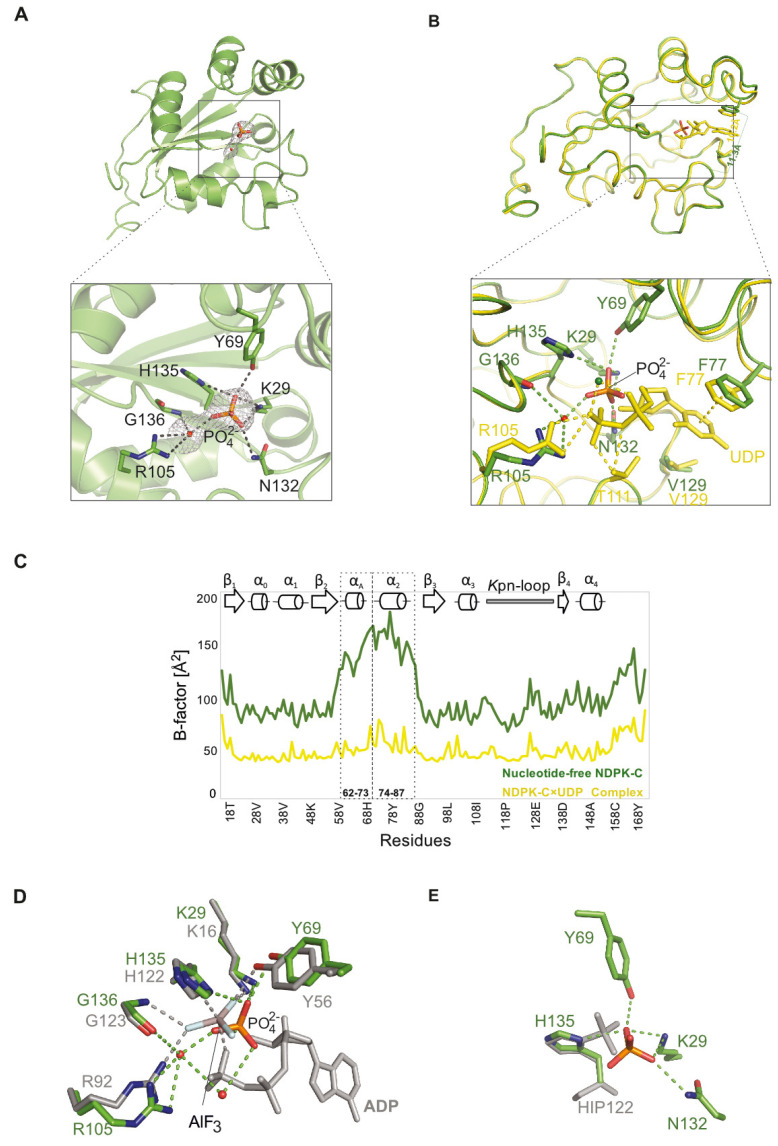
Structure of nucleotide-depleted human NDPK-C (**A**) Ribbon model of one nucleotide-depleted NDPK-C subunit (green); only protomer 1 of the hexameric structure is shown for clarity. The Fo-Fc electron density map of the phosphate ion and water molecule contoured at σ = 1.3 is shown as a grey mesh together with the modeled HPO_4_^2−^ in orange sticks. The phosphate is stabilized by interaction with Lys29, Tyr69, Asn132, His135 and a water molecule that is linked to Arg105 and Gly136. (**B**) Superposition of nucleotide-depleted (green) with one NDPK-C×UDP subunit (yellow) shows the movement of helix α2 (including Ala60 to Arg73) in the presence and absence of UDP. The distance between Val129 (located in Kpn-loop) and Phe77 (located in helix α2), which can be considered as a gate to access the active site, is increased from 10.2 Å in the NDPK-C×UDP complex to 11.3 Å in depleted-UDP NDPK-C. An enlarged view of NDPK-C active site in UDP (yellow) and nucleotide-free structures (green) shows that the free phosphate ion is localized in the same area as α-phosphate in NDPK-C×UDP structure but closer to His135 to establish direct interaction. (**C**) The value of the B-factor in the UDP-depleted structure is increased in both helices (αA and α2) compared with NDPK-C in complex with UDP. Increasing the B-factor in this area resembles the movement of the respective region in depleted-nucleotide NDPK-C (green plot). (**D**) Superposition of nucleotide-depleted NDPK-C (green) and NDPK from *D. discoideum* (grey 1KDN). The bound AlF_3_ forms an accurate analogue of the transition of γ-phosphate in NDPK×ADP×AlF_3_ complex from *D. discoideum* (grey). The superposition of nucleotide-depleted NDPK-C (green) and NDPK from *D. discoideum* (grey 1KDN) shows the phosphate ion in depleted NDPK-C is positioned in a very similar place as AlF3 in NDPK from *D. discoideum*, which can be viewed as a transition state of NDPK-C. Water molecules are shown as red spheres. (**E**) Superposition of depleted NDPK-C (green) and NDPK from *D. discoideum* (grey 1NSP) shows the similar localization of phosphate in depleted-nucleotide NDPK-C as phosphate, which is covalently bound to the catalytic histidine in *D. discoideum* NDPK.

**Figure 5 ijms-25-09768-f005:**
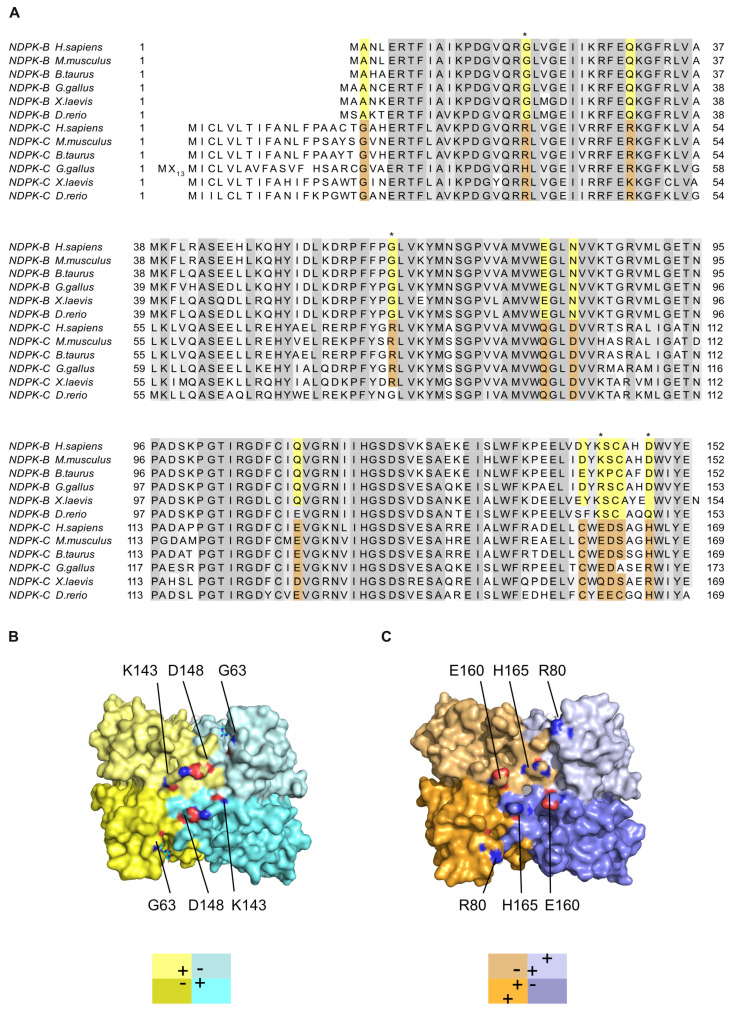
NDPK-B and -C isoforms show differences in surface potential—(**A**) Primary sequence alignment of NDPK-B and -C orthologues from the indicated species. Conserved residues are shaded in gray. Residues which differ between the isoforms but are specifically conserved within one isoform are highlighted in yellow and brown, respectively. Residue positions that (i) show isoform-specific conservation; (ii) are located on the surface of the hexameric 3D structure; and (iii) are chemically different in terms of their isoforms are indicated with stars: Gly63/Arg80, Asp148/His165 and Lys143/Glu160. (**B**) Surface display of NDPK-B (PDB 1NUE) with conserved isoform-specific residues labeled using atomic coloring. (**C**) Surface view of NDPK-C with conserved residues of Arg80, His165 and Glu160 highlighted on the surface. The two isoforms show clear differences in the location of charged residues on their surface.

**Table 1 ijms-25-09768-t001:** Data collection and refinement statistics. Values in parentheses correspond to the outer resolution shell.

PDB Entry	8QW3	8QVZ	8QW0	8QW1	8QW2	8QVY
Ligands	ADP	cAMP	GDP	ADP, Mg^2+^	UDP, Mg^2+^	None
Data Collection						
X-ray source	ESRF ID23-1	ESRF ID30A-3	ESRF ID30A-3	ESRF ID23-2	ESRF ID23-1	ESRF ID30A-3
Wavelength (Å)	1.005	0.9677	0.9677	0.8731	0.6888	0.9677
Detector	PILATUS 6M-F	EIGER X 4M	EIGER X 4M	PILATUS3 X 2M	EIGER2 X CdTe 16M	EIGER X 4M
Space group	*P*1	*P*1	*P*2_1_	*C*2	*C*2	*C*2
Unit cell dimensions						
*a* (Å)	52.32	52.02	104.3	106.5	106.5	106.2
*b* (Å)	67.87	67.41	78.64	116.3	116.0	115.5
*c* (Å)	68.42	68.15	112.4	84.07	83.84	82.61
*α* (°)	108.5	108.7	90	90	90	90
*β* (°)	112.3	112.4	97.56	93.05	93.28	94.32
*γ* (°)	101.1	100.9	90	90	90	90
*Vm* (Å^3^/Da)	1.89	1.86	2.17	2.47	2.46	2.40
Solvent content (%)	35.1	34.0	43.5	50.3	50.0	48.9
Resolution range (Å)	60.1–1.25 (1.28–1.25)	45.0–1.77 (1.81–1.77)	46.6–2.17 (2.23–2.17)	42.0–2.10 (2.15–2.10)	47.7–1.87 (1.94–1.87)	47.3–2.64 (2.71–2.64)
*R_merge_* (%)	4.9 (94.7)	12.7 (101.4)	15.2 (138.8)	18.4 (150.9)	4.2 (118.7)	12.7 (239.0)
<*I*/σ(*I*)>	8.6 (0.7)	6.2 (1.2)	9.6 (1.0)	10.0 (1.7)	15.0 (1.1)	9.45 (0.9)
CC_1/2_ (%)	97.0 (40.1)	99.3 (61.2)	99.8 (54.9)	99.5 (46.6)	99.7 (69.7)	99.2 (48.0)
Data completeness (%)	94.2 (89.0)	97.3 (88.4)	99.5 (94.2)	95.8 (73.1)	99.9 (99.5)	96.4 (98.2)
Average redundancy	2.8 (1.8)	4.6 (4.4)	10.3 (7.2)	6.6 (5.9)	5.2 (4.9)	7.3 (7.4)
Wilson *B* (Å^2^)	12.8	20.8	46.2	41.6	44.0	79.1
Refinement						
Max. resolution (Å)	1.25	1.77	2.17	2.10	1.87	2.64
Total nr. reflections	198,505	71,473	94,493	57,087	83,594	28,215
Test set	2119	2091	1996	2382	2091	1263
*R_work_* (%)	14.7	16.5	18.9	17.2	17.0	19.7
*R_free_* (%)	17.5	21.0	22.9	20.5	20.3	24.7
Protein atoms	7425	7481	14,483	7284	7300	7073
Other non-solvent atoms	177	147	346	168	156	30
Solvent atoms	542	408	303	399	313	0
RMSD bond lengths (Å)	0.0116	0.0106	0.0112	0.0091	0.0092	0.0073
RMSD bond angles (°)	1.671	1.664	1.812	1.641	1.618	1.470
Ramachandran						
favored (%)	98	98	98	98	98	97
allowed (%)	2	2	2	2	2	3
outliers (%)	0	0	0	0	0	0
Average *B* (Å^2^)						
protein	21.3	26.8	50.8	41.4	61.0	103.3
other non-solvent	27.9	30.1	66.9	51.0	77.4	133.1
solvent	30.9	30.1	41.1	41.8	57.3	-

**Table 2 ijms-25-09768-t002:** PDB entries and crystallization conditions.

PDB Entry	8QW3	8QVZ	8QW0	8QW1	8QW2
Ligands	ADP	cAMP	GDP	ADP, Mg^2+^	UDP, Mg^2+^
Nucleotide concentration	1 mM	5 mM	3 mM	1 mM	2 mM
Space group	*P*1	*P*1	*P*2_1_	*C*2	*C*2
Crystallization condition	6% PEG 3350			7% PEG 8000	
	150 mM Li_2_SO_4_			27% Glycerol	
	100 mM Tri-sodium citrate pH 5.4			40 mM KH_2_PO_4_	

## Data Availability

All models will be made available on the Protein Databank (PDB) with the following accession codes: 8QW3, 8QVZ, 8QW0, 8QW1, 8QW2 and 8QVY.

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
