# Peer review of "Mechanistic Insights into Substrate Recognition of Human Nucleoside Diphosphate Kinase C Based on Nucleotide-Induced Structural Changes"

_ijms, 2024, doi:10.3390/ijms25189768_

Round 1

Reviewer 1 Report

Comments and Suggestions for Authors

General comments: 

The authors of this study have elucidated a series of structures of NDPK C in complex with different ligands, which enriches our understanding of the NDPK family. Additionally, by integrating previous structural and biochemical experimental data, the authors have proposed several hypotheses regarding the catalytic mechanism of NDPK. 

Specific comments:

1. P3 lane128 'Interactions between two subunits...also part of the dimer interface' In Figure 1C and 1D, the three-dimensional structure does not have a strong sense of depth. A zoom-in view should be added to display detailed information of the interaction interface.

2. Figure 1.A, B already effectively demonstrate the assembly of the NDPK hexamer; panel E is redundant and should be removed.

3. Figure 1.F Check the units on the vertical axis.

4. A comparison of overall conformation should be made for the magnesium-free and magnesium-containing ADP complexes.

5. What is the conformation and location of D138 in the magnesium-free ADP complex? Can it be shown in Figure 2.B?

6. Figure 2.A&C show the map density for the ligand ADP. Why not show the map density for the ligands GDP, UDP, and cAMP as well?

7. P10 lane 283-285 'Superposition of human homo-hexameric...human NDPK-C (data not shown)'. An additional figure is needed.

8. P14 lane 409 'A variety of structures for NDPK from various species have been solved.' References could be added.

Author Response

In the following we will address the comments of the reviewers individually

Reviewer 1

The authors of this study have elucidated a series of structures of NDPK C in complex with different ligands, which enriches our understanding of the NDPK family. Additionally, by integrating previous structural and biochemical experimental data, the authors have proposed several hypotheses regarding the catalytic mechanism of NDPK.

We thank the reviewer for the appreciation of our work and her/his constructive suggestions.

Specific comments:

  1. P3 lane 128 'Interactions between two subunits...also part of the dimer interface' In Figure 1C and 1D, the three-dimensional structure does not have a strong sense of depth. A zoom-in view should be added to display detailed information of the interaction interface.

Thank you for pointing that. The zoom-in view has been added (Fig. 1E)

  1. Figure 1.A, B already effectively demonstrate the assembly of the NDPK hexamer; panel E is redundant and should be removed.

We agree. The scheme (panel E) has been removed and replaced with the zoom in view (see comment 1)

  1. Figure 1.F Check the units on the vertical axis.

Thanks for pointing that out. Units were shifted to the left to be clearly visible.

  1. A comparison of overall conformation should be made for the magnesium-free and magnesium-containing ADP complexes.

We provide a superposition of the magnesium-free and magnesium-containing ADP complex as a supplemental Fig. S2.

  1. What is the conformation and location of D138 in the magnesium-free ADP complex? Can it be shown in Figure 2.B?

D138 in the magnesium-free ADP complex is now shown in Fig. 2B

  1. Figure 2.A&C show the map density for the ligand ADP. Why not show the map density for the ligands GDP, UDP, and cAMP as well?

Thank you for this suggestion. Map densities have been added for GDP (Fig. 3B), UDP (Fig. 3C), and cAMP (Fig. 3D)

  1. P10 lane 283-285 'Superposition of human homo-hexameric...human NDPK-C (data not shown)'. An additional figure is needed.

Thank you for pointing that out. The new figure is found as supplemental Fig. S3.

  1. P14 lane 409 'A variety of structures for NDPK from various species have been solved.' References could be added.

We agree. Three relevant reviews are cited now.

Reviewer 2 Report

Comments and Suggestions for Authors

In the manuscript “High-resolution structures of human nucleoside diphosphate kinase C in complex with different nucleotides” Amjadi et al reported high resolution X-ray structures of Nucleoside diphosphate kinases-C bound to various nucleotides. They compared the obtained 3D structures with that of the other NDPK isoforms published by others. DEspite the interesting system, I have a few concerns and need further clarifications to consider this manuscript to be published in IJMS.

  1. In the abstract it is not clear which method was used to determine the structures. It is better to specify X-ray crystallography.

  2. Authors claim in the manuscript that they determined ‘novel’ structures of the NDPK-C isoform. While going through the manuscript I do not understand what is the novelty here. The authors compared the structures with other well studied isoforms and it is discussed as comparable to other published structures. If there is a novelty in the new structures it is not well described. The new structures are also hexameric as reported in the case of other NDPK variants. What is the novelty here? It is misleading and authors should refrain from making unnecessary statements to make the manuscript interesting.

  3. Why is NDPK-C a dimer and then trimer of the dimers? What is the physiological significance? What is the physiological relevance of a hexameric conformation?

  4. In the case of apo NDPK-C, did the authors try crystallizing without phosphate ions? I am referring to the section 2.3, page #11.

  5. Hexameric structure: I understand the hexameric structure obtained is also supported by SEC-MALS. Is there any significance if the protein exists as a monomer? Is the hexameric state formed due to the buffer composition used for crystallization? 

  6. Did the authors rule out the existence of monomers by other methods such as NMR? A simple 1D NMR can tell whether protein is hexameric or monomeric. Diffusion NMR is another option to assess the size of the protein. Then if needed isotopic labeling and multidimensional NMR measurements can be performed to support the argument. I understand that the group is not focusing on NMR research and please note that this is a constructive suggestion to find further novelty in the structure.

Author Response

In the following we will address the comments of the reviewers individually

Reviewer 2:

 In the manuscript “High-resolution structures of human nucleoside diphosphate kinase C in complex with different nucleotides” Amjadi et al reported high resolution X-ray structures of Nucleoside diphosphate kinases-C bound to various nucleotides. They compared the obtained 3D structures with that of the other NDPK isoforms published by others. DEspite the interesting system, I have a few concerns and need further clarifications to consider this manuscript to be published in IJMS.

We thank the reviewer for her/his interest in our work and will address all her/his concerns in our responses and the revised manuscript.

In the abstract it is not clear which method was used to determine the structures. It is better to specify X-ray crystallography.

Thank you for that comment we state now X-ray crystallography specifically in the abstract.

Authors claim in the manuscript that they determined ‘novel’ structures of the NDPK-C isoform. While going through the manuscript I do not understand what is the novelty here. The authors compared the structures with other well studied isoforms and it is discussed as comparable to other published structures. If there is a novelty in the new structures it is not well described. The new structures are also hexameric as reported in the case of other NDPK variants. What is the novelty here? It is misleading and authors should refrain from making unnecessary statements to make the manuscript interesting.

The reviewer asks for the novelty in our structures compared to the other NDPK class 1 isoforms, especially to the structurally well described hexamers of NDPK-A and NDPK-B. In contrast to these isoforms, no crystal structure of NDPK C has been published. The reason for this lack likely lies in the higher hydrophobicity of NDPK-C and higher instability in aqueous solution. That is for us the most feasible reason, why only a preliminary 3D structure which is for example not able to resolve the nucleotide binding pocket has been deposited in 2005 but not published yet.

Although the hexameric structure we have obtained has a high degree of similarity in the monomer folding as well as the general assembly of the hexamer when compared to NDPK-A or NDPK-B, there are for example differences in the surface potential (see Fig. 5) which might be of relevance for interactions with other NDPK isoforms and other partners in protein complexes. Especially the very C-terminal region of NDPK exhibits low conservation compared to the other human isoforms. This region locates on the surface of the hexameric complex and makes the surface specific for the C-isoform.  This region may be involved in formation of higher complexes also including heteroproteins and may be critical for isoform-specific recognition. We made this more evident in Fig. 5 of the revised manuscript.

So far a careful comparison of different nucleotide complexes was carried out only by Nguyen et al. (Nguyen S et al, FEBS J. 2021, doi: 10.1111/febs.15607. PMID: 33089641) using the NDPK from the fungus Aspergillus fumigatus. However, an equivalent study is lacking for the human isoform NDPK-C. Due to its evolving role in heart disease, knowledge of the detailed binding mode of NDPK-C to various nucleotide ligands is a prerequisite for future drug development. In contrast to previous publications, we additionally addressed the structures with different bound nucleotides (substrates and inhibitors) as well as without bound nucleotide thoroughly leading to data in high resolution of the nucleotide binding pocket. The novelty lies here in the sum of the details. To make that more evident to the readers we changed the title to “Mechanistic insights into substrate recognition of human nucleoside diphosphate kinase C based on nucleotide induced structural changes”.

We discuss the recent biological and pathophysiological evidence which detected novel specific functions in mitochondria and at the plasma membrane, which are to our understanding “not unnecessary statements to make the manuscript interesting”. From our own research we are aware that important features of NDPK heterooligmerization, e.g. NDPK-B/C oligomers are specifically interacting with heterotrimeric G protein βγ-dimers (summarized in Abu-Taha et al., 2017). NDPK-C is the mediator of the interaction and the NDPK-B/C complex additionally acts as protein histidine kinase. To understand the role of NDPK-C in this complex and also how NDPKs might act as protein histidine kinase high quality structural data are needed. Interestingly, we obtained data, indicating a higher flexibility and the possibility of an opening of the cleft containing the intermediately phosphorylated catalytic His residue in the absence of a bound nucleotide. These data are also structurally novel and might be important with regard to the protein histidine kinase activity.  

.

Why is NDPK-C a dimer and then trimer of the dimers? What is the physiological significance? What is the physiological relevance of a hexameric conformation?

Thank you for these questions. Based on evolutionary and functional data obtained by many researchers what is described here is the general assumption how class 1 NDPKs assemble and function. The current knowledge has been nicely summarized in the review by Georgescauld et al., 2020,:

“All functional NDPKs so far described in the literature are oligomers [27,37,46]. Eukaryotic

NDPKs are hexamers, while prokaryotic NDPKs are hexamers or tetramers [6,46]. Regardless of the oligomerization state of these enzymes, the oligomers are always constructed from a dimer formed by two subunits assembled head to tail [6,46]….”

“Functional NDPK oligomers are formed from the assembly of dimers [6,46]. The dimers

self-assemble, two or three times into tetramers or hexamers [22]. One main difference between hexameric and tetrameric NDPKs comes from the C-terminal segment (Figure 1), which is longer for most hexameric NDPKs: it brings the two dimer subunits into contact and also participates in their trimerization….”

“Only after assembly into hexamers do the Kpn/α0 subdomains get structured and the NDPK becomes enzymatically functional….”

“In the NDPK from D. discoideum the P100S mutation combined with the successive deletion of the last 5 C-terminal amino acids leads to a loss of activity as well as the dissociation of the enzyme into dimers [48,49]…”

Taken together, at least for the class 1 NDPKs, the physiological active form in eukaryotic cells is the hexamer. It is formed out of three dimers assembled head to tail, as described also in the manuscript. We agree with the reviewer that the text in the original manuscript might have been confusing. To make this section clearer to the readers we modified the corresponding text:

The NDPK family members are structurally highly conserved from bacteria to humans and all eukaryotic and archaebacterial NDPKs are hexameric. Most of the prokaryotic NDPKs are however tetramers. Functional oligomers are always constructed from a dimer formed by two monomer subunits assembled head to tail. The enzymatic active and physiological functional hexameric NDPKs are built of three dimers…

In the case of apo NDPK-C, did the authors try crystallizing without phosphate ions? I am referring to the section 2.3, page #11.

Thank you for the question. Yes, we tried to crystallize NDPK-C in the absence of a nucleotide in cocktails including pH stabilizing buffers others than phosphate. But under these conditions, no crystals grew and we observed protein precipitation. Thus, our attempts to reveal a structure without phosphate ions failed.

Hexameric structure: I understand the hexameric structure obtained is also supported by SEC-MALS. Is there any significance if the protein exists as a monomer? Is the hexameric state formed due to the buffer composition used for crystallization?

As already pointed out above, the hexamer is the natural existing and physiological relevant form of eukaryotic class 1 isoform. The vast majority of the purified recombinant protein is already assembled as hexamer when it is purified from E. coli. To make this more evident we double checked our SEC-MALS data at elution volume corresponding to lower molecular mass proteins but could not see any hints for dimers in solution. In addition, we added Fig. S1 in which the molecular masses of His-tagged NDPK-C and also of NDPK-A and NDPK-B has been analyzed by mass photometry before SEC. In all three preparations the hexamers are predominantly detected. Higher aggregates, most likely consisting of two hexamers, can be found, too, to a small extent. Most likely, this is artificial and has to be attributed to the high protein concentrations obtained during protein purification.

Did the authors rule out the existence of monomers by other methods such as NMR? A simple 1D NMR can tell whether protein is hexameric or monomeric. Diffusion NMR is another option to assess the size of the protein. Then if needed isotopic labeling and multidimensional NMR measurements can be performed to support the argument. I understand that the group is not focusing on NMR research and please note that this is a constructive suggestion to find further novelty in the structure.

Thank you for that interesting suggestion. We agree that 1D NMR would be another valid method to assess the oligomeric state of the protein. As pointed out above already, the prevalent formation of the hexamer in solution has been proven by SEC-MALS which is fully supported by the added mass photometry data (Fig. S1). A detailed NMR analysis including proposed isotope labeling and multidimensional NMR spectroscopy would indeed be a good idea for a separate study.

Round 2

Reviewer 2 Report

Comments and Suggestions for Authors

I appreciate the effort of the authors to incorporate my suggestions and further explanation to my questions. By including the suggested modifications, the authors have addressed my major concerns and suggestions. 

However, as I expressed my concern regarding the novelty in the presented structures I do not think it has the utmost importance to be novel. Except in the abstract the novelty is not well addressed. I understand the structures presented have various features and are important in the context of previous studies and existing structures. Despite the explanation of the authors to address my comment I still believe it is better not to mention the ‘novelty’, especially in the abstract. I assume the authors will modify it accordingly.

Other than that, this is now a reasonably good article and I support the publication of this manuscript in IJMS, after the necessary modifications.

Author Response

In the following we will address the remaining comment of reviewer 2:

"I appreciate the effort of the authors to incorporate my suggestions and further explanation to my questions. By including the suggested modifications, the authors have addressed my major concerns and suggestions."

Thank you very much for your appreciation of our work. Your comments have been very helpful.

"However, as I expressed my concern regarding the novelty in the presented structures I do not think it has the utmost importance to be novel. Except in the abstract the novelty is not well addressed. I understand the structures presented have various features and are important in the context of previous studies and existing structures. Despite the explanation of the authors to address my comment I still believe it is better not to mention the ‘novelty’, especially in the abstract. I assume the authors will modify it accordingly."

 Thank you for pointing that out. We agree and therefore replaced " the novel straucture" by "these high resolution structures" in the abstract.

Other than that, this is now a reasonably good article and I support the publication of this manuscript in IJMS, after the necessary modifications.

Thank you very much for this positive assessment.